# Observation and simulation of neutral air density in the middle atmosphere during the 2021 sudden stratospheric warming event

Junfeng Yang[1,2,*], Jianmei Wang[1,2], Dan Liu[1,2], Wenjie Guo[1,2], Yiming Zhang[1,2]

[1]State Key Laboratory of Space Weather, National Space Science Center, Chinese Academy of Sciences, Beijing, 100190, China;

[2]Key Laboratory of Science and Technology on Environmental Space Situation Awareness, Chinese Academy of Sciences, Beijing, 100190, China

*Correspondence to*: Junfeng Yang (yangjunfeng@nssc.ac.cn)

**Abstract.** Sudden stratospheric warmings (SSWs) are dramatic events in the polar winter stratosphere that are accompanied by atmospheric parameter anomalies in the stratosphere and mesosphere. Microwave Limb Sounder and Global Navigation Satellite System Occultation Sounder observations on board the Chinese FengYun 3 satellites indicate a rapid increase of over 50% in the mesospheric density at high latitudes around the onset date during the 2021 major SSW event. The amplification of the zonal mean density around the onset is proportional to the latitude increase with a maximum increment of 83.3% at 59 km above 80 °N, which is more than three times larger than the climatological standard deviation (23.1%). The horizontal density distributions are influenced by the changing polar vortex fields. A simulation using a specified dynamics version of the Whole Atmosphere Community Climate Model is consistent overall with the observations and presents a severe change in the planetary wave forcing and residual meridional circulation mass flux followed by a change in the density tendency. These results demonstrate that the observed enhanced density is primarily attributed to the altered planetary waves and residual circulation during the SSW event. The observations and simulations also indicate that the density anomalies could extend to middle latitudes. Obvious density disturbances in the upper stratosphere and mesosphere were observed by lidar deployed in Beijing (40.3 °N, 116.2 °E).

## 1 Introduction

For the near space vehicles flighting at the airspace 20–100 km and the space shuttle, entry phase of which begins at an altitude of 122 km and ends at the ground, the atmosphere variations will highly influence specific fuel consumption, engine performance, drag, communication and flight control (Weaver et al., 2011; Chen et al., 2023). The density variations are manifested as density shear, differences from the standard atmosphere model, and density perturbations dependence on longitude, latitude, and season (Hale et al., 2002). The significant difference between the actual atmosphere density and widely used standard upper air models, such as the 1976 US reference atmosphere, has often been found and contributed to attack angle bias from the reference angle of attack profile, causing adverse thermal consequences (Champion, 1990; Hale et al., 2002). On several flight experiments, the atmospheric drag, a direct function of the neutral atmospheric density at a given

altitude, has varied by up to 19% over a few seconds. It is indicated that middle atmospheric density variations require additional attention from aircraft designers and the aircraft industry (Hale et al., 2002; Weaver et al., 2011). Furthermore, 38 Starlink satellites were destroyed by a unexpected geomagnetic storm that led to a density enhancement of over 20% at ~210 km and a larger atmospheric drag on February 4, 2022 (Dang et al., 2022). Atmospheric density is not only important for Low-Earth Orbit satellites at the altitudes (160-1600 km), but also for the vehicles in the middle atmosphere at the altitudes (20-100 km).

Sudden stratospheric warmings (SSWs) are dramatic events that occur approximately once every 2 years in the winter stratosphere over the North Pole and occasionally over the South Pole (Lee et al., 2021; Sherhag, 1952; Davis et al., 2022; Rao et al., 2020). During SSW events, there is a sudden warming of 40–60 K in the polar stratospheric temperature over a few days and a deceleration of the stratospheric zonal wind, which may even reverse from climatological eastward to westward (Andrews et al., 1987). Most observations and simulations have confirmed that SSW events could have profound effects not only on the stratosphere but also on the troposphere, mesosphere, thermosphere, and even ionosphere (Kodera et al., 2016; Yang et al., 2017; He et al., 2020; Liu et al., 2019; Oberheide et al., 2020; Davis et al., 2022). Kodera et al. (2016) determined the relationship between the stratospheric vortex and tropospheric circulation during an SSW event. Yang et al. (2017) investigated the zonal wind response derived from medium frequency radar observations over Langfang, China (39.4 °N, 116.7 °E), to the 2013 major SSW event. Yuan et al. (2012) reported the winds and temperature in the mesopause region (80–102 km) during the 2009 SSW event using the Colorado State University sodium Doppler wind–temperature lidar at Fort Collins (41 °N, 105 °W). Liu et al. (2019) presented a remarkable perturbation of the total electron content and the deviation of the total electron content in the ionosphere during the 2018 major SSW event. Taking advantage of the global coverage of observational data from satellite instruments, Manney et al. (2009) demonstrated the transport of trace gases from the upper troposphere to the lower mesosphere using the Aura Microwave Limb Sounder (MLS) during the 2006 major SSW event. Oberheide et al. (2020) reported O/N2 column density depletion of more than 10% at the onset of SSW using observations by the Global-Scale Observations of the Limb and Disk instrument on the geostationary SES-14 communications satellite during the 2019 major SSW event. The atmospheric responses observed by ground sites always have a longitudinal asymmetry, which is caused by the structure and evolution of the planetary waves (PWs) and the pole vortex during SSW events (Chandran and Collins, 2014).

In general, atmospheric variations during SSW events in most observational and modeling investigations are given in the form of variations at a certain pressure level or the approximate altitude. The atmospheric density is always neglected because it is a diagnostic parameter in most observational and model data, resulting in the density evolution during SSW events rarely being reported. It is reported that a dramatically varying actual atmospheric density often contributes to a angle of attack bias, which can have adverse thermal consequences (Hale et al., 2002). Therefore, SSW events, as the most spectacular global atmospheric phenomenon, should not be ignored and require recognition, given their effects on the neutral air density, which may highly affect the atmospheric drag, flight control, engine performance, and specific fuel consumption of aircraft.

In early January 2021, a major SSW event occurred and this was an unusually prolonged major warming event (Lu et al., 2021; Davis et al., 2022). Our work explores the temporal evolution and horizontal distributions of the neutral air density

during the January 2021 SSW event based on observations from the MLS and Global Navigation Satellite System (GNSS) Occultation Sounder (GNOS) on board the Chinese FengYun 3 (FY-3) satellites and Beijing (40.3 °N, 116.2 °E) lidar. A specified dynamics version of the Whole Atmosphere Community Climate Model (SD-WACCM) is used to diagnose the driving mechanism and reveal the detailed processes of the air mass transport. This paper is organized as follows. In Sect. 2, the datasets and the model used in the study are described. In Sect. 3, the observation and simulation results are presented.

Finally, a discussion and conclusion are given in Sect. 4.

## 2 Data and methods

### 2.1 Satellite datasets: Aura/MLS and FY-3C/GNOS

The Aura satellite was launched in July 2004 by the National Aeronautics and Space Administration (NASA) and operates in a sun-synchronous orbit with 98 inclinations at an altitude of 705 km (Waters et al., 2006). The orbital period of Aura is

approximately 1.7 h, circling the Earth 13–14 times per day. The MLS on board the Aura satellite measures millimeter- and submillimeter-wavelength thermal emission from the limb of Earth's atmosphere. Each scan takes 24.7 s. The data coverage of the MLS product is from latitudes of 82 °S to 82 °N on each orbit, with an along-track horizontal resolution of 200–300 km. The accuracy of the temperature retrievals has been validated by Schwartz et al. (2008) and Livesey et al. (2015). The atmospheric temperature and geopotential height (GPH) profiles were obtained with approximately 3300–3600 measurements

80    per day, a vertical coverage of 261–0.001 hPa (10–92 km), and a vertical resolution of approximately 1.3 km below 50 km and 5–8 km above 50 km (Wu and Eckermann, 2008). The version 4.2x (level 2) of the temperature ($T$) and GPH ($H$) at pressure levels ($P$) from the MLS data were used to calculate the atmospheric air density ($\rho$). The geometric height ($Z_g$) was derived from the GPH ($H$) using the following equation based on the WGS-84 geoid (World Meteorological Organization, 2018):

$$Z_g = \frac{H r(\varphi) g_0}{r(\varphi) g(\varphi) - H g_0} , \tag{1}$$

where $g_0$ is defined as 9.80665 m/s and $r(\varphi)$ is the effective radius of the Earth for a given latitude ($\varphi$), where

$$r(\varphi) = 6378137/(1.006803 - 0.006706 * sin(\varphi)^2). \tag{2}$$

$g(\varphi)$ is the normal gravity on the surface of an ellipsoid of revolution, calculated as

$$g(\varphi) = 9.780325 * \left( \frac{1 + 0.00193185 * sin(\varphi)^2}{(1 - 0.00669435 * sin(\varphi)^2)^{0.5}} \right). \tag{3}$$

Then, the geometric height ($Z_g$) for the height above sea level ($Z_s$) becomes

$Z_s = Z_g + Z_e ,$                      (4)

where $Z_e$ is the geoid height calculated using the EGM96 Geopotential Model. The density ($\rho$) is determined via the ideal gas law:

$\rho = P/RT,$                                               (5)

where $R$ is the gas constant derived from the NRLMSISE-00 Atmosphere Model (Cheng et al., 2020). Finally, the density/pressure and temperature were interpolated logarithmically and linearly to an altitude resolution of 1 km. The global distribution of atmospheric density was gridded in the region of $5°×20°$ (latitude $×$ longitude) using the daily data.

The GNSS GNOS is a new-generation payload on board the Chinese FY-3 series of operational meteorological satellites launched in September 2013 (Sun et al., 2018). FY-3C/GNOS is designed to observe radio occultation data using GNSS signals from both the Chinese BeiDou Navigation Satellite System and the US Global Positioning System. Currently, the FY-3C/GNOS data products have been validated and are widely used for atmosphere scientific applications. FY-3C/GNOS gives the temperature and pressure data at altitudes above sea level. The atmospheric density can then be directly calculated using the ideal gas law.

Since the atmospheric density decreases exponentially with altitude, relative density deviation was used to represent the density change. The relative density deviation $\delta$ at each altitude is calculated as:

$$\delta = \frac{\rho_o - \rho_r}{\rho_r} \times 100\%, \tag{6}$$

where $\rho_o$ is the observed density and $\rho_r$ is the reference density, which may differs in different figures and will be described in each figure.

## 2.2 Ground dataset: Beijing Rayleigh lidar

The Rayleigh lidar deployed in Beijing (40.3 °N, 116.2 °E) is part of the lidar chain in the Meridian Project, which is a space environment monitoring system measuring the space environment from the ground (Wang, 2010). The lidar emits pulsed laser, which elastically collides with atmospheric molecules. The received backward Rayleigh scattering signal can be captured by the detector of the lidar system and is used to calculate the atmospheric density and temperature (Yue et al., 2014). In this paper, the temperature and density profiles from 30 km to 70 km were retrieved from lidar observational data from December 2020 to February 2021.

## 2.3 Reanalysis data: MERRA-2

The Modern Era Retrospective-analysis for Research and Applications, version 2 (MERRA-2) was used to diagnose the 2021 SSW event. The dataset consists of the latest atmospheric reanalysis of the modern satellite era developed by the Global Modeling and Assimilation Office of NASA; this reanalysis is described by Gelaro et al. (2017). MERRA-2 is produced with version 5.12.4 of the Goddard Earth Observing System atmospheric data assimilation system. The reanalysis has an approximate resolution of $0.5° × 0.625°$ (latitude $×$ longitude) and 72 vertical levels from the surface to 0.01 hPa (~78 km). The temperatures, horizontal winds, and GPHs provided by MERRA-2 are used to analyze the evolution of the zonal mean temperature, zonal wind, and planetary waves during SSW events.

## 2.4 SD-WACCM model simulation

WACCM is a component of the Community Earth System Model developed by the National Center for Atmospheric Research (Garcia et al., 2007; Richter et al., 2010; Garcia et al., 2019). WACCM uses a specified dynamics (SD) run, called SD-WACCM, with a horizontal resolution of $1.9\,°\times2.5\,°$ (latitude $\times$ longitude) to simulate the global atmospheric view. MERRA-2 data, described above, are used to constrain the WACCM temperature and horizontal winds in the troposphere and stratosphere. The nudging factor is 10% from the surface to 50 km and then decreases to 0 at 60 km. The model is free-running above 60 km. It has been shown in previous reports that model simulations can serve as comprehensive tools to reproduce the atmospheric evolution of SSW events (De Wit et al., 2014; Chandran and Collins, 2014; Lee et al., 2021; Yang et al., 2017). The output of SD-WACCM includes the temperatures and GPHs, and the air densities are calculated using the same method as for the Aura/MLS data.

## 2.5 Method of calculating the residual meridional circulation

To analyze the changes in the air motion, the residual meridional circulation (RMC) is calculated in this study in the context of the transformed Eulerian mean approach (Andrews et al., 1987). The meridional and vertical components of the RMC within the transformed Eulerian mean approach can be calculated using the formulas described by Andrews et al. (1987) and Gille et al. (1987):

$$\overline{v}^* = \overline{v} - \frac{1}{\rho}\frac{\partial}{\partial z}\left(\rho\frac{\overline{v'\theta'}}{\partial\overline{\theta}/\partial z}\right), \tag{7}$$

$$\overline{w}^* = \overline{w} + \frac{1}{r\cos\varphi}\frac{\partial}{\partial\varphi}\left(\frac{\cos\varphi\overline{v'\theta'}}{\partial\overline{\theta}/\partial z}\right), \tag{8}$$

where $v$ and $w$ are the zonal mean meridional and vertical velocities, respectively; $\rho$ is the background neutral air density; $z$ is the altitude; $\theta$ is the potential temperature; $\varphi$ is the latitude; and r is the Earth's radius. In addition, the overbars denote the zonal mean values and the primes indicate the deviations of the hydrodynamic quantities from their zonal mean values, where $v' = v - \overline{v}$, $\theta' = \theta - \overline{\theta}$, and $\overline{v}^*$ and $\overline{w}^*$ represent the residual meridional and vertical components, respectively. $\overline{w}^*$ is considered to be proportional to the net rate of adiabatic heating, which roughly influences the temperature. Upward $\overline{w}^*$ leads to cooling, and downward $\overline{w}^*$ leads to heating (Gille et al., 1987). The residual meridional circulation approximates the movements of atmospheric air masses and the transport of gas tracers.

## 3 Results

### 3.1 Rapid increase in density observed by Aura/MLS and FY-3C/GNOS

Figure 1a and 1b shows the temporal evolution of the temperature gradient and the zonal wind from December 1, 2020, to February 28, 2021, based on the MERRA-2 data. The zonal mean temperature gradient between the North Pole and $60\,°$N increases by close to 45 K over a week, reverses on January 2, and peaks on January 4. The zonal mean zonal wind over $60\,°$

N at 10 hPa decreases in later December 2020 and turns from eastward to westward until January 5, 2021. According to the World Meteorological Organization criterion of a major SSW event (Kodera et al., 2016), that is, simultaneous reverses in the zonal mean temperature gradient and the zonal mean zonal winds, the onset date is specified as January 5, 2021, which is indicated by vertical black lines in Fig. 1. Although the westward winds after the onset are discontinuous and interrupted by eastward winds, they are considered to occur during a single major SSW event (Lee, 2021). Lu et al. (2021) reported that the zonal mean zonal wind in the upper stratosphere reversed from eastward to westward around December 31, 2020, and that the westward anomalies lasted until February 16, 2021, for a total duration of approximately 53 days.

The temporal evolution of the zonal mean temperature and density over 70 °N observed by Aura/MLS and FY-3C/GNOS data from December 2020 to February 2021 is shown in Fig. 1c–1f. The latitude window calculated according to the zonal mean is set to 67.5–72.5 °N, which is filled with approximately 100 observed profiles from Aura/MLS and 37 profiles from FY-3C/GNOS. The dashed lines in Fig. 1e and 1f indicate the maximum temperatures at the stratopause altitude. The stratopause over 70 °N falls from 50–60 km in early December 2020 to 45 km at 270 K on January 2, 2021, while the mesospheric temperature decreases to below 190 K. A colder stratopause is maintained at approximately 50 km during the SSW event and becomes warmer and higher in early February. In Fig. 1f, the value is shown as the relative deviation between the density during the SSW event and the climatological average winter density computed by the Aura/MLS statistics during the period from 2004 to 2021. It is impressive that a density increment as high as 50% occurs on January 4, even though atmospheric disturbances are expected during an SSW event, such as the four other peaks on December 13 and 18, January 15, and February 1. The former two peak density anomalies are short-lasting at only 1–2 days while the latter two last 3–4 days in the mesosphere and the positive upper stratospheric density lasts the entire SSW period. The density and temperature gradually return to the climatology state after February 16.

To validate the intense anomalies during the 2021 major SSW event, the stratospheric temperature and density evolution given by the FY-3C/GNOS data is shown in Fig. 1e and 1f, with the white gaps indicating the absence of observations. The temperature at 40 km increases suddenly to 274 K on January 4, 30 K warmer than that in December 2020. Subsequently, a cooler temperature above 45 km and a warmer temperature below 45 km is maintained until mid-February. There are some differences in the stratospheric density values between Fig. 1f and 1d that may be due to their different reference profiles compared with the winter average density for FY-3C/GNOS and the climatological average density for Aura/MLS, in addition to their different detection precisions. The FY-3C/GNOS density shows a sharp enhancement on January 9 that is nonexistent in the Aura/MLS data. However, the FY-3C/GNOS data indicate a rapid density increase around the onset and a higher density during the SSW event; this is generally consistent with the Aura/MLS data.

According to the observed temporal evolution of the density, the latitude–altitude density structures were investigated using the Aura/MLS measurements for three different stages of the 2021 SSW event. Figure 2 gives the relative deviation of the global density with the global mean density for November 20, 2020 (pre-SSW); January 4, 2021 (around the onset date); and February 1, 2021 (after the onset date). The climatological mean density were obtained from the 18-year observational data of Aura/MLS during the period of 2004–2021. In the climatological situation, the density over the equator is higher than the

density of the other regions at 20-25 km, and the density over the Southern Hemisphere is overall higher than the density over the Northern Hemisphere at 25-90 km. The global distribution on 20 December 2020 is generally consistent with climatic winter feature. On 4 January 2021, the atmospheric density over the Arctic region at 30-85 km has increased by more than 20% larger than the global mean density, and this increase is relatively significant compared to the climatic Standard deviations. On 7 January 2021, the atmospheric density over the Arctic region at 50-90 km quickly returns to that situation before the onset, while atmospheric density at the 30-50 km is still higher. The maximum increase located at 46 km is also more significant compared to the climatic Standard deviations.

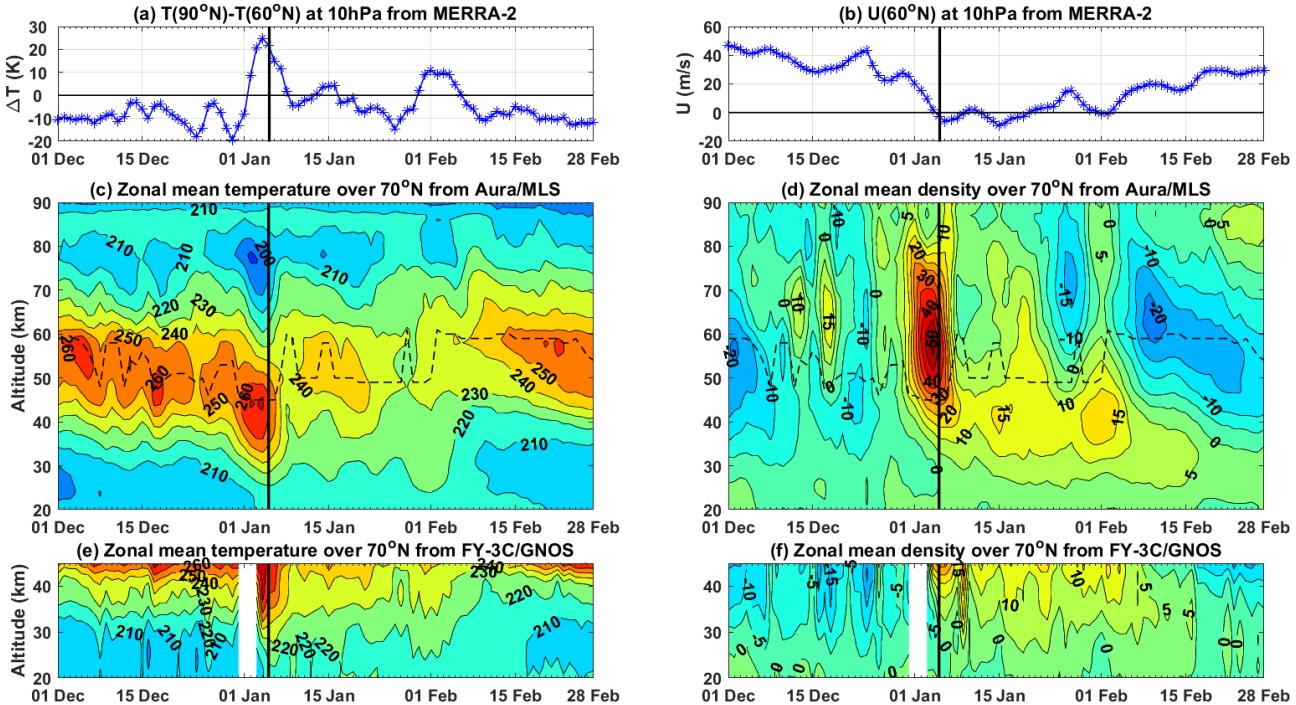

**Figure 1: Temporal evolution of the temperature, zonal wind, and density from December 1, 2020, to February 28, 2021. (a) Zonal mean temperature gradient between the North Pole and 60 °N and (b) zonal mean zonal winds over 60 °N from the MERRA-2 data. (c) Zonal temperature and (d) the relative density deviation between that during the SSW event and the climatology over 70 °N from the MLS data. (e) Zonal temperature and (f) the relative density deviation between that during the SSW event and the winter average (December, January, and February) over 70 °N from the FY-3C/GNOS data. The vertical black lines indicate the onset date, and the dashed lines in panels (e) and (f) indicate the maximum temperatures as the stratopause altitude.**

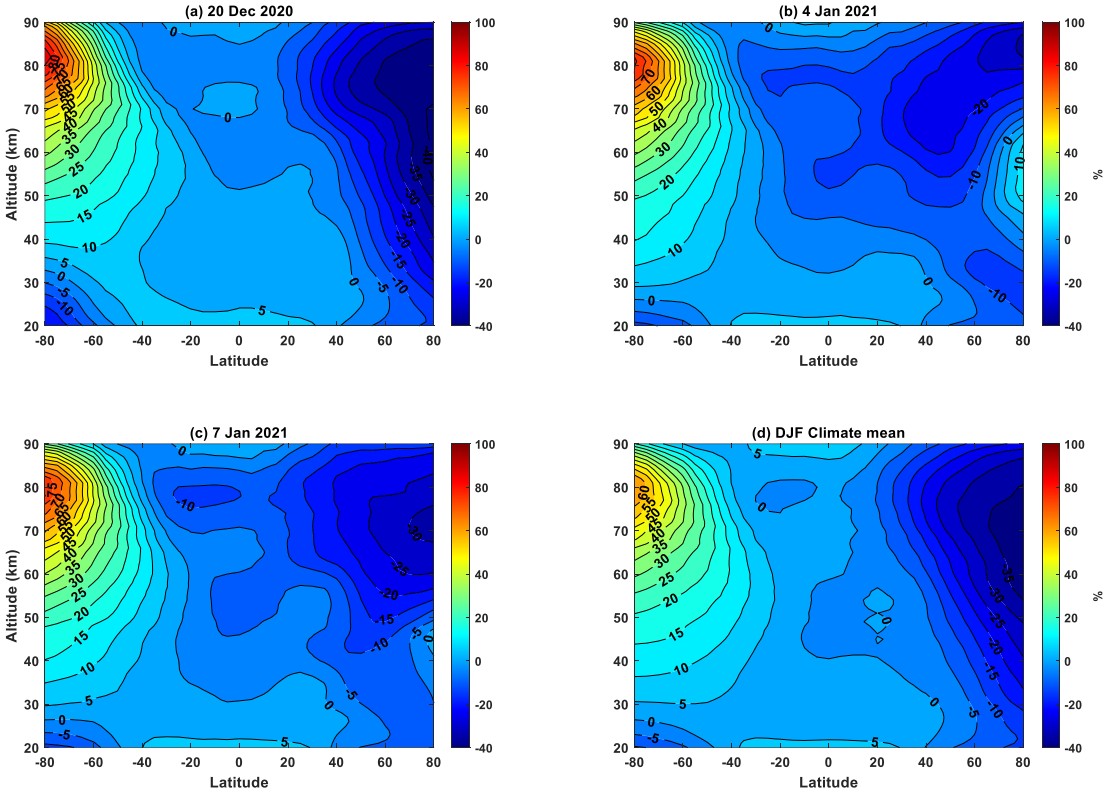

**Figure 2: Relative deviations between the zonal mean density and the global mean density: (a) November 20, 2020; (b) January 4, 2021; and (c) January 7, 2021 from Aura/MLS data and (d) climatic winter average obtained for the period from 2004 to 2021.**

To test whether the density change during the 2021 major SSW is significant, the relative density deviations from the climatological mean situation are compared with the climatological standard deviation shown in Fig. 3. The climatological standard deviation was also obtained from the 18-year observational data of Aura/MLS during the period of 2004–2021. Figure 2d demonstrates that the largest standard deviations are centred at 58 km over the North Pole and this may be primarily due to the frequent planetary waves in the extratropical winter stratosphere and the lower mesosphere (Smith, 2012). The major differences on November 20, 2020 (Fig. 3a), are lower than the climatological standard deviations, except in a very small region at 90 km over high latitudes in the Southern Hemisphere, indicating that the current atmospheric condition is similar to the common climatological state. The density anomalies with large enhancements in the stratosphere and mesosphere over high latitudes in the Northern Hemisphere are particularly noticeable around the onset date in Fig. 3b. The amplification of the zonal mean density is enlarged with increasing latitude. The maximum increment was 83.3% at 59 km over 80 °N, which is more than 3 times larger than the climatological standard deviation (23.1%). The zonal mean positive relative deviation region can extend to ~50 °N in the stratosphere and even lower latitudes in the upper atmosphere. In addition, there are negative

relative deviations over the middle and low latitudes. The weakened density in the mesosphere is lower than −10%, which exceeds the climatological standard deviation. On January 7, the density in the mesosphere returned to the climatology state; this is earlier than the return in the stratosphere. The peak density amplification altitude during the event fell to 46 km over high latitudes.

Longitudinal variations in the temperatures and wind have been reported at the same latitude, modulated by planetary waves 220 and gravity waves (Hoffmann et al., 2007; Chandran and Collins, 2014; Yang et al., 2017). Figure 4 reveals the climatological horizontal density distributions at altitudes of 80 km, 60 km, and 40 km for November 20, 2020, and January 4 and 7, 2021, from Aura/MLS. In the climatological state, the North Pole is occupied by a low-pressure vortex and the standard density deviations increase with latitude in a ring form located on the North Pole. On November 20, 2020, the polar vortex at 40 km starts to move as the result of an anticyclone with high pressure that developed over the Aleutian Islands. Prior to the SSW 225 event, the difference inside the vortex was negative at each level and the pressure of most regions was within the climatological standard deviations. Around the onset, the North Pole was wholly invaded by the anticyclone when the vortex was displaced and asymmetrically split. Simultaneously, the density increased at the North Pole with a maximum of 93.5%, far greater than the climatological standard deviations. On January 7, the density anomalies weakened when the polar vortex was gradually restored and the anticyclone faded at each level. Figure 4 shows that the positive anomalies of the density were confined to the 230 anticyclone and the negative anomalies were in the cyclone, indicating that the horizontal density distribution was modulated by planetary waves during the SSW event.

The location of the lidar deployed in Beijing (40.3 °N, 116.2 °E) is indicated by the red asterisks in Fig. 4. Around the onset, the Beijing lidar was in the split polar vortex, demonstrating that the above atmosphere was influenced by the SSW event. Figure 5 shows the daily temperature and density over Beijing from the lidar and that from the Aura/MLS data. Lidar operation 235 is permitted only on cloudless nights, resulting in many data gaps. However, the evolution of the temperature and density observed by the lidar is consistent with those observed by Aura/MLS. It is interesting that the stratospheric temperature over Beijing does not increase as rapidly as that at high latitudes, but rather decreases, and the maximum temperature is observed at a very high altitude (55 km for the lidar and 62 km for Aura/MLS) on January 2. The positive density anomalies of more than 10% at 45–60 km on December 15 and 29 and January 9 observed by Aura/MLS were also captured by the Beijing lidar. 240 Although the deep density drop at the onset was unfortunately missed, the disturbances over Beijing are inferred to have been controlled by the everchanging vortexes during the SSW event. Furthermore, as the lidar density data is a directed parameter by Beijing lidar system, the consistency between lidar observations with the Aura/MLS observations confirms the computational methods described in Sect. 2.

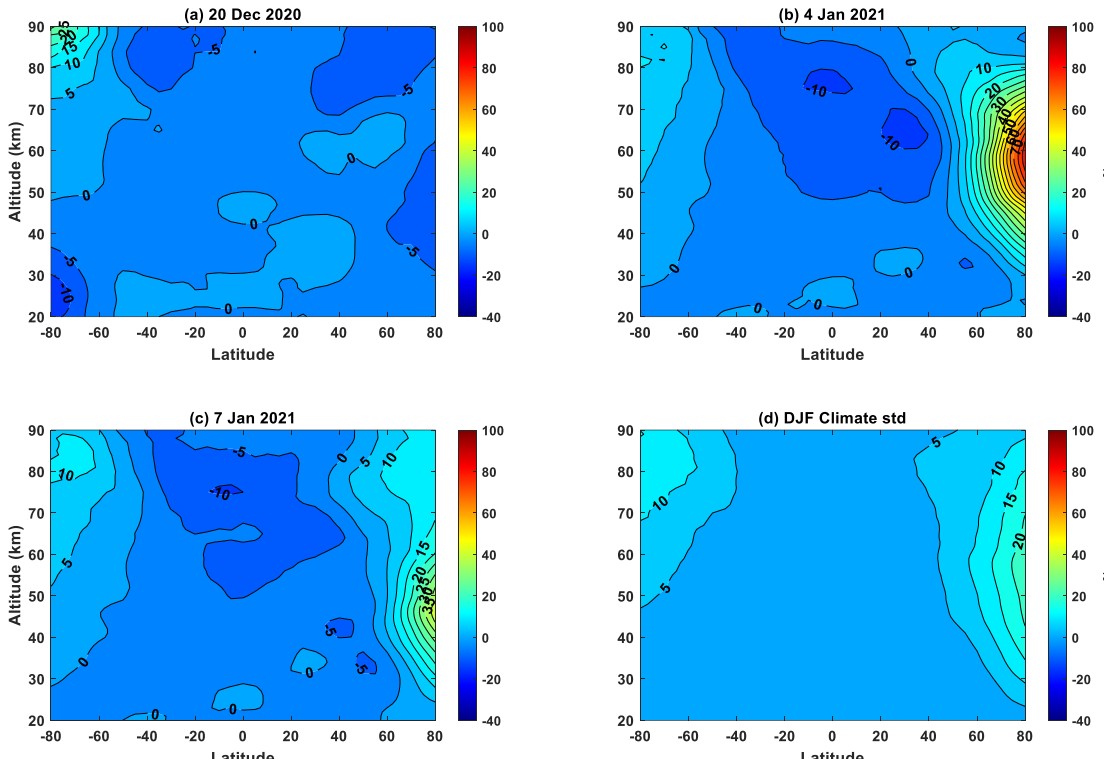

**Figure 3: Relative deviations between the zonal mean density and the climatological average density: (a) November 20, 2020; (b) January 4, 2021; and (c) January 7, 2021 from Aura/MLS data. (d) The standard deviations of the zonal mean density for the winter average obtained for the period from 2004 to 2021.**

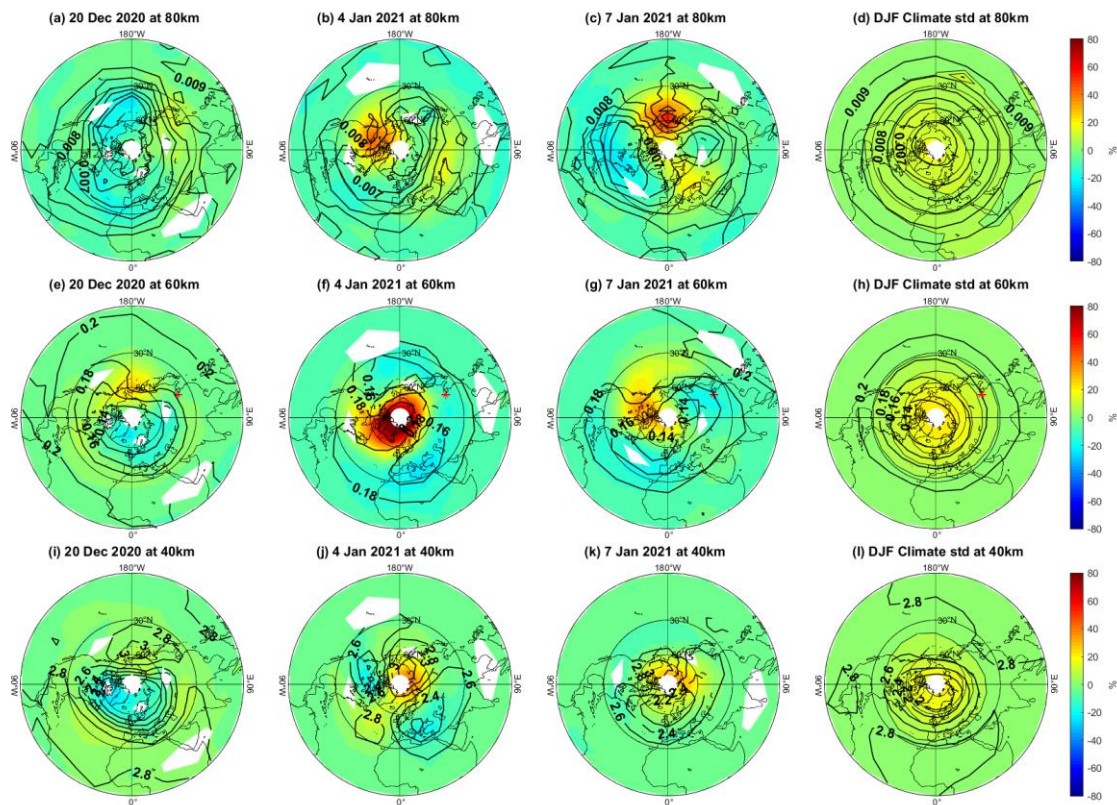

**Figure 4: Horizontal density distributions, values of which are shown as relative deviations between the density and the climatological mean density, at (a)–(c) 80 km, (e)–(g) 60 km, and (i)–(k) 40 km for November 20, 2020 (first column), January 4, 2021 (second column), and January 7 (third column) from Aura/MLS measurements. Standard deviations of the daily zonal mean density for the winter average are obtained for the period from 2004 to 2021 at (d) 80 km, (h) 60 km, and (l) 40 km. The overlaid black contours indicate the pressures. The red asterisks indicate the location of the Beijing lidar. Stereographic projections are used in the maps with 0 °longitude at the bottom and 90 °E to the right, and the domain is from 0 °to 90 °N.**

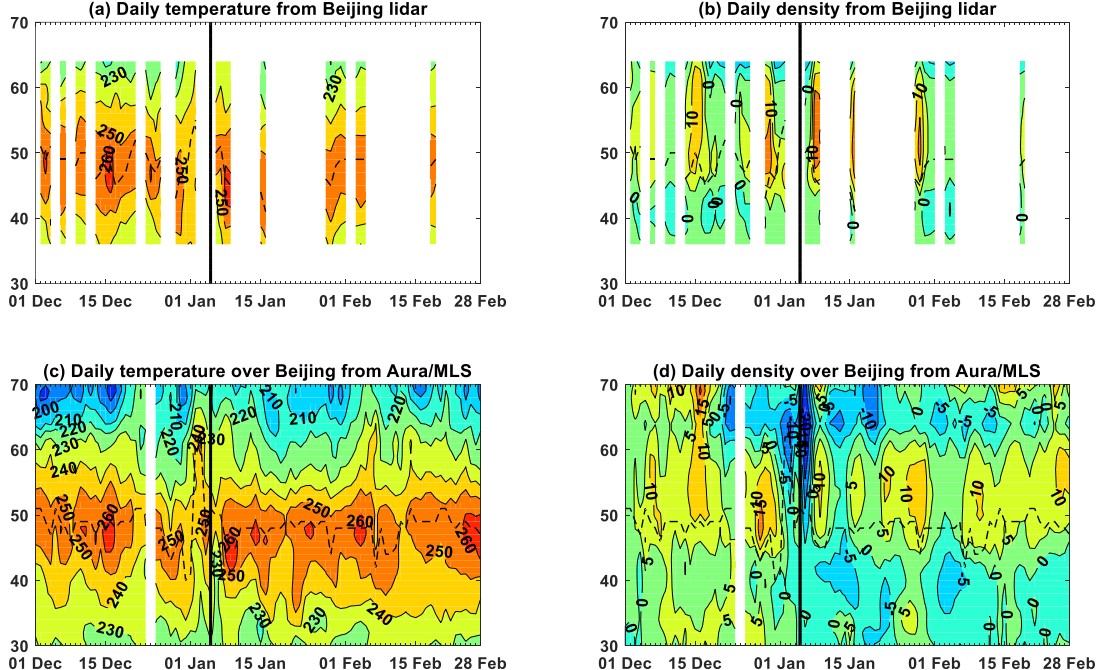

**Figure 5: Temporal evolution of the temperature and relative density deviation over Beijing from (a)–(b) the lidar and (c)–(d) Aura/MLS data. The reference density in (b) and (d) is 2020–2021 winter average density from lidar and climatological winter average density from Aura/MLS, respectively. The vertical black lines indicate the onset date, and the dashed lines indicate the maximum temperatures as the stratopause altitude.**

### 3.2 Planetary waves and residual meridional circulation

The global circulation during the 2021 major SSW event was reproduced using SD-WACCM simulation. The simulated temperature and density, as shown in Fig. 6a–b, agree with the observations, excepting some small differences in detail within the topmost altitudes. The mesopause temperature is slightly colder than the observations around the onset date, and a decrease in the density appears above 80 km. According to the ideal gas equation, a lighter density is expected when the temperature is warmer; that is not in accordance with the increased stratospheric density accompanied by a warming temperature. In addition, the cooling magnitude of approximately 20 K (roughly -8% of pre-existing temperature) at 58 km cannot lead to a density enhancement of above 50%, assuming other conditions are stable. This indicates that the density variation is possibly forced by atmospheric dynamics. The observed changing polar vortex during the SSW event has been illustrated in Fig. 4, and Figure 6c and 6d present the amplitudes of the planetary wavenumber 1 (PW1) and planetary wavenumber 2 (PW2) simulated by SD-WACCM. Throughout the entire winter, PW1 is stronger overall than PW2 before and after the onset; however, PW2 only appears around the onset. The five density peaks previously mentioned in Fig. 1d correspond to different planetary waves.

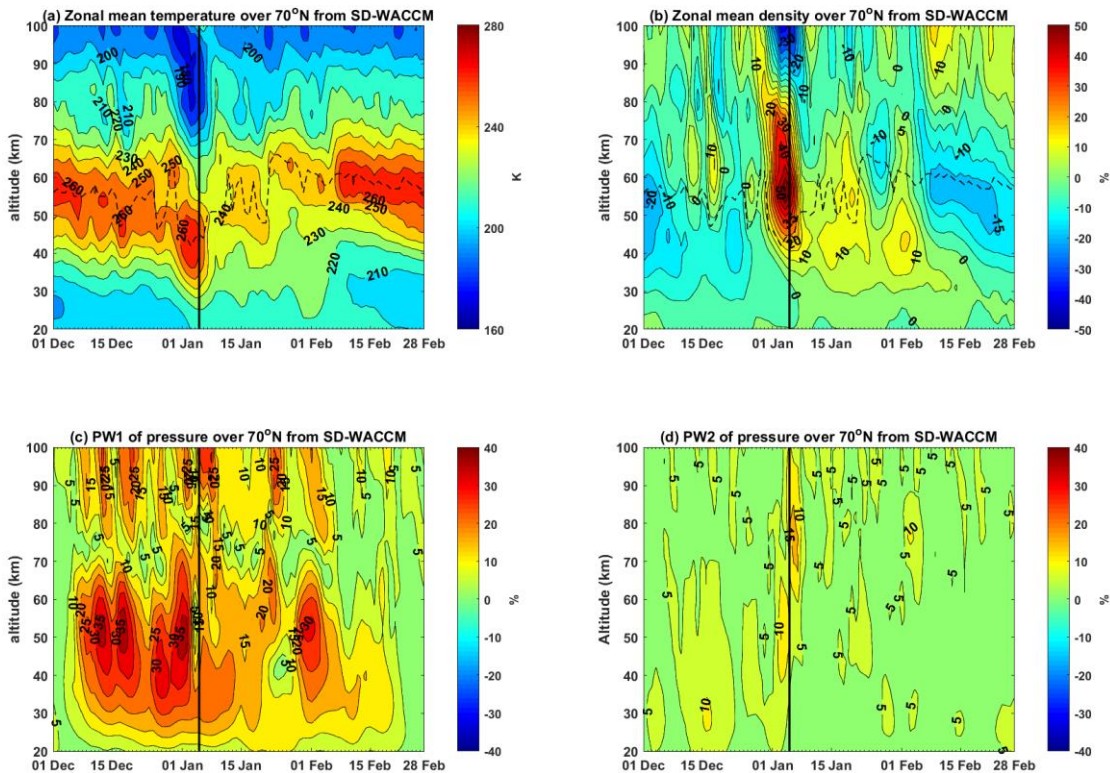

**Figure 6: Temporal evolutions over 70 °N simulated by SD-WACCM: (a) The zonal mean temperature. (b) The zonal mean density. (c and d) The amplitudes of planetary wavenumber 1 (PW1), and planetary wavenumber 2 (PW2) of the pressure. The vertical black lines indicate the onset date, and the dashed lines in panels (a) and (b) indicate the maximum temperatures as the stratopause altitude.**

Figure 7a shows the temporal evolution of the Eliassen–Palm flux divergence at 56 km over the Northern Hemisphere simulated by SD-WACCM. The planetary waves produce periodic eastward forcing alternated with westward forcing at 56 km over high latitudes in the Northern Hemisphere prior to the SSW event and then a persistently westward forcing around the SSW onset. This noticeable change in the planetary wave (PW) forcing parallels a notably altered residual circulation with enhanced poleward motion at the same altitude (Fig. 7b). Figure 8 demonstrates the latitude–altitude distributions of RMC at different stages of the 2021 major SSW event. The general structures of the residual circulation on November 5, 2020, prior to the SSW event, are similar to the climatological winter characteristics, that is, there are two symmetrical circulations with opposite directions below 37 km with a clockwise circulation in the Northern Hemisphere, an anticlockwise in the Southern Hemisphere, and a global clockwise residual circulation above 37 km. On January 1, 2021, the strengthened poleward motion in the upper stratosphere accelerates the clockwise circulation below 60 km in the Northern Hemisphere and reverses the RMC above 60 km from clockwise to anticlockwise.

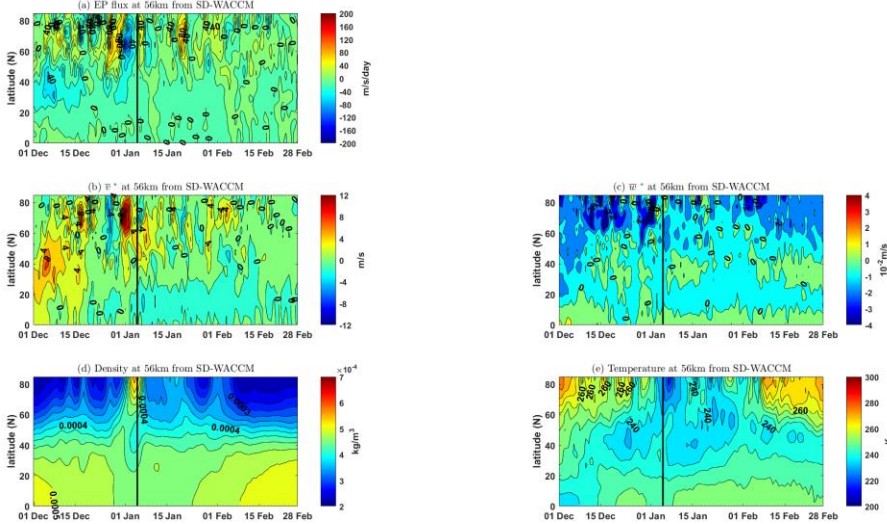

**Figure 7: Temporal evolution at 56 km over the Northern Hemisphere simulated by SD-WACCM of the (a) Eliassen–Palm (EP) flux, (b) residual meridional velocity, (c) residual vertical velocity, (d) density, and (e) temperature. The vertical black lines indicate the onset date.**

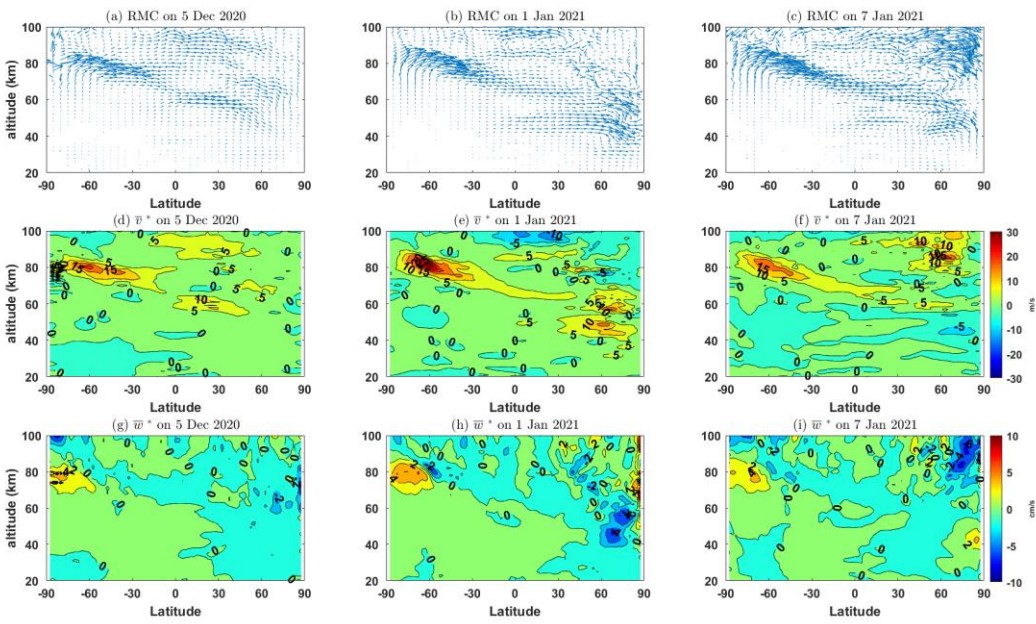

**Figure 8: Latitude–altitude distributions of the residual meridional circulation (RMC) simulated by SD-WACCM. (a)–(c) RMC vectors, (d)–(f) residual meridional velocity, and (g)–(i) vertical velocity over 70 °N on November 5, 2020 (right column), (b) January 1, 2021 (middle column), and February 7, 2021 (left column).**

Following the changes in the PW forcing and the residual circulation, there is a particular change in the density distribution from the situation in which the density over lower latitudes is larger than that over high latitudes in December 2020 to a reversal gradient with latitude around the onset (Fig. 7d). The increased PW westward forcing and the poleward motion also exhibit a wide extent to 25 °N in Fig. 7a and 7b; this arrangement may transport more air masses from the low and middle latitudes with high density to the North Pole with low density, gradually causing the density decrease over the low and middle latitudes seen in Fig. 7d. This process would explain the rapid density enhancement over 70 °N and the decrease over middle latitudes seen in Fig. 1d.

As in many previous investigations, there is a downward motion below 60 km and a upward motion above 60 km over high latitudes, as shown in Fig. 8h; these motions produce adiabatic heating of the stratosphere and cooling of the mesosphere and lower thermosphere, respectively. Although the vertical velocities at 56 km are weaker than those at other altitudes (Fig. 8h), the vertical velocity (Fig. 7c) is related to the temperature (Fig. 7e). The stronger cold temperature over high latitudes and the wide-range cold temperature over middle latitudes are accompanied by positive vertical velocities.

On January 7, 2021, during the recovery stage after the SSW onset date shown in Fig. 8c, 8f, and 8g, the RMC turns to an opposite direction in the Northern Hemisphere. The stratospheric northward motion is weakened and even turns southward, transporting atmosphere air masses from high latitudes to middle and low latitudes, weakening the prior drastic increase in the density. Because the southward movement is intermittent and has a weaker strength, the density recovery process has a longer period and spans the entire SSW event (Fig. 7d).

The RMC mass fluxes were calculated by multiplying the atmospheric density by the residual meridional and vertical velocities, such that $F_m = \rho \overline{v}^*$ and $F_v = \rho \overline{w}^*$ (Koval et al., 2021). Figure 9 shows the temporal evolutions of the RMC mass flux and the daily density tendency over 70 °N. There is a clear relationship between the RMC and the density. Significantly, the altered RMC mass flux around the onset lasts for a long duration of approximately 2 weeks, corresponding to the density accumulation and recovery over 70 °N. In the simulation, a reversed meridional mass flux appears above 80 km and causes a density decrease (Fig. 6b) that is not observed by Aura/MLS (Fig. 1d); this could be due to the limited detection height and resolution of Aura/MLS. Apart from the acute change around the SSW onset, there are multiple occurrences of altered RMC mass fluxes followed by density increases that last only approximately 1–3 days, such as on December 12 and 17. Vertical mass fluxes with different directions in the stratosphere and mesosphere during each RMC change could transport air to lower and higher altitudes. In general, the SD-WACCM simulation perfectly reproduces the rapid density increase during the 2021 major SSW event and usefully demonstrates the corresponding relationship between the altered planetary waves and the RMC.

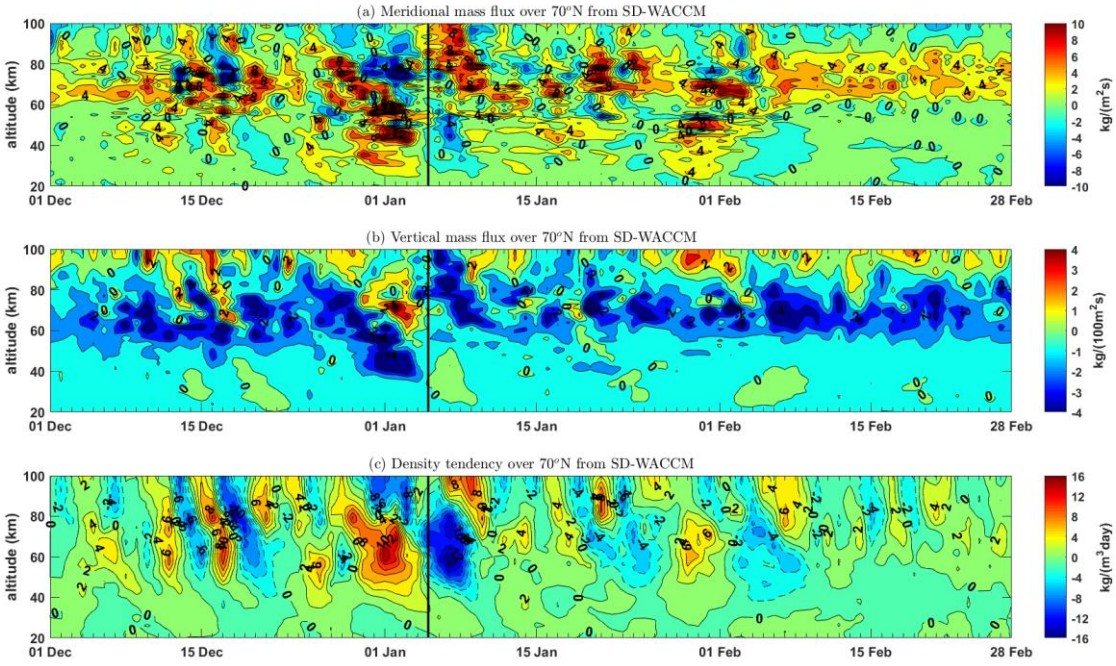

**Figure 9: Temporal evolution of the (a) meridional mass flux, (b) vertical mass flux, and (c) daily tendency of the zonal mean density simulated by SD-WACCM over 70 °N. Solid and dashed lines indicate positive and negative values, respectively, indicating density increases and decreases, respectively. The mass flux is scaled by exp⟨/⟩(-z/7) at each altitude.**

## 4 Discussion and conclusions

The neutral atmospheric density at a given altitude is important in the design drag of aircraft and planning aircraft trajectories and flight conditions. However, most previous studies concerning SSW events, which are the most dramatic events in the polar winter stratosphere, have only revealed the stratospheric temperature warming in the form of variations at a certain pressure level or an approximate altitude. However, the atmospheric density, as an implicit parameter, is always neglected and rarely reported. According to the ideal gas equation, warm and cold temperatures correspond to lighter and heavier densities, respectively. However, a rapid density increase in the middle atmosphere over high latitudes during the 2021 major SSW event was observed by Aura/MLS and FY-3C/GNOS. Figure 10 shows the temporal evolutions of the temperature and density at the isobaric levels, and the corresponding approximate heights are indicated in the right ordinate. This isobaric-level form result is commonly used at present (Manney et al., 2009; Chandran and Collins, 2014; Kodera et al., 2016; Lu et al., 2021). It can be seen from the Fig. 10 that the stratospheric temperature warming around the onset date causes the density to decrease by -14.33% at ~ 43km, and the mesospheric temperature cooling causes the density to increase by 14.42% at 75 km. This density change at the isobaric levels can be considered as the results caused by the change in thermodynamic temperature change, which is under the same pressure. Supposing that the atmosphere is stationary, the temperature cooling magnitude of only

approximately 20 K (-8%) at 58 km could not cause the density enhancement of more than 50% recorded in the observational data. These huge differences between this density evolution at the isobaric levels (Fig. 10) and the observation density

evolution at altitudes (Fig. 1), indicate that the density increment as high as 50% around the onset data is mainly caused by atmospheric dynamics (Fig. 6 and 7). Stober et al. (2012) has observed the phase shift between the temperature and the density at approximately 89 km several days prior to the 2009/2010 SSW event using a Juliusruh MR (54.6 °N, 13.4 °E). They indicated that the density variation was likely caused by planetary waves. On the other hand, most previous studies only give the atmospheric temperature changes like Fig. 10a, not the density and pressure, which are unable to provide accurate density

information to the aircraft designers and the aircraft industry. Since the atmospheric drag calculated on flight experiments should considers the density at a given altitude, the altitude results in our work are very meaningful. This large density increase should cause more attention in the calculation of atmospheric drag.

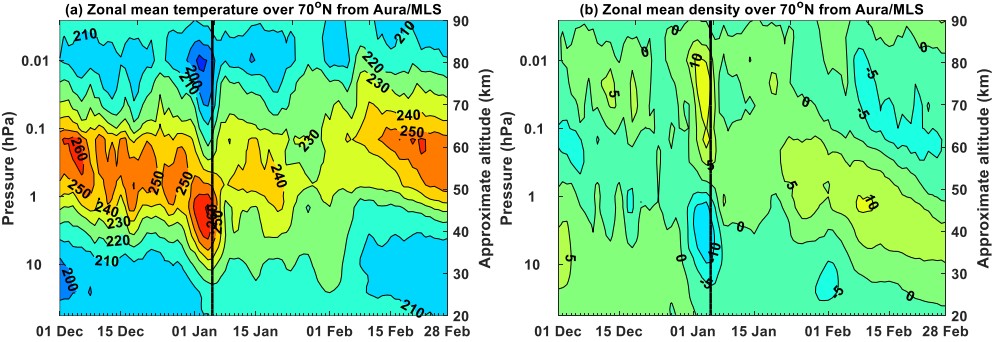

**Figure 10: Temporal evolution of the temperature and density from December 1, 2020, to February 28, 2021. (a) Zonal temperature**
**and (d) the density deviation between that during the SSW event and the climatology over 70 °N from the MLS data. The vertical black lines indicate the onset date.**

The global atmospheric simulation by the SD-WACCM model is overall consistent with the observations and provides a further insight into the relationship between the density evolution and the planetary waves and residual circulation. Around the onset of the 2021 major SSW event, the stratospheric PWs forcing turned westward and the clockwise RMC is divided into

two circulations with opposite directions below and above 60 km in the Northern Hemisphere. In climatological winter, the density decreased with increasing latitude (Fig. 2d). The accelerated northward motion from higher density to lower density could transport more air masses from low and middle latitudes to the North Pole, resulting in the density over the North Pole being larger than that over lower latitudes around the SSW onset (Fig. 7d). The changes in the RMC around the onset seen in Fig. 8 last much longer than the other changes at other times. Therefore, the rapid density increase with a maximum of more

than 50% in the Aura/MLS and FY-3C/GNOS observations is thought to have been caused by the severe change in the planetary waves and the residual circulation.

The observations and simulations also show that the effects of SSW extended to the middle and low latitudes. Obvious density disturbances in the upper stratosphere and mesosphere were observed by a lidar deployed in Beijing (40.3 °N, 116.2 °E) in agreement with the Aura/MLS observations. The influences of the simulated PW forcing and RMC also extend to lower

latitudes, as shown in Fig. 6. Oberheide et al. (2020) presented an observed $O/N_2$ column density between latitudes of $\pm 60\,°$ that was depleted by more than 10% during the 2019 January SSW event. Their investigation indicates that the $O/N_2$ depletion was not caused by geomagnetic activity but rather by enhanced global wave driving and associated residual circulation changes in the lower thermosphere. A similar situation is seen in Fig. 2b and 6d with the density decrease over low latitudes caused by the SSW event. However, it is should be noted that planetary waves are more dominant in the middle atmosphere, while tidal

waves rather than planetary waves would contribute more in the lower thermosphere (Liu et al., 2010). In light of the demonstrated important role of PWs and the RMC in the transportation of air masses and chemical substances, our work could provide a better understanding of the neutral air density evolution in the middle atmosphere during dramatic SSW events and is therefore useful to aircraft designers and the aircraft industry.

According to the classification based on the stratospheric response (Kodera et al., 2016), 2022 major SSW is more of a mixed

type of warming event. We have also investigated two other typical style SSW. The 2006 major SSW was a typical displacement type event triggered by the PW1 (Chandran and Collins, 2014), while the 2018 major SSW was a split type event by PW2 (Rao et al., 2018). The sudden positive anomalies of neutral air density are also observed around the onset date in the 2018 and 2006 major SSW. In 2018, the maximum value of 49.54% at 48 km occurs on 15 February lags 4 days after the onset date. In 2006, the maximum value of 51.44% at 61 km occurs on 11 January, 11 days before the onset date. Their difference

detail needs more explorations in future work.

**Code and data availability.** The Aura/MLS data are available at https://acdisc.gesdisc.eosdis.nasa.gov/data/Aura_MLS_Level2. The FY-3C/GNOS data can be obtained from the National Satellite Meteorological Center, China Meteorological Administration, at http://satellite.nsmc.org.cn. The Beijing lidar data were obtained from the Chinese Meridian Project Database (https://data.meridianproject.ac.cn/). WACCM is a component

model for the atmosphere in the Community Earth System Model developed at the National Center for Atmospheric Research (NCAR), and the source code is available at https://www.cesm.ucar.edu/models/releases. The MERRA-2 forcing data in SD-WACCM were obtained at https://rda.ucar.edu/datasets/ds313.3.

**Author contributions.** JY proposed the scientific ideas, and wrote most of the paper. WG and JW provided data processing of Aura/MLS data and lidar. JY and JW contributed to data SD-WACCM simulation. DL and YZ validated the manuscript.

**Competing interests.** The authors declare that they have no conflict of interest.

**Disclaimer.** Publisher's note: Copernicus Publications remains neutral with regard to jurisdictional claims made in the text, published maps, institutional affiliations, or any other geographical representation in this paper. While Copernicus Publications makes every effort to include appropriate place names, the final responsibilitylies with the authors.

**Acknowledgements.** Thanks to the Goddard Earth Sciences Data and Information Services Center for providing the

400 Aura/MLS data. Thanks to the the National Satellite Meteorological Center of China Meteorological Administration and the Chinese Meridian Project Database for providing the FY-3C/GNOS data. Thanks to the Chinese Meridian Project Database for providing the lidar data. Thanks to the NCAR for providing WACCM model. Thanks to the Chinese Meridian Project

Database for providing the lidar data. Thanks to the Global Modeling and Assimilation Office for producing the MERRA-2 data.

**Financial support.** This research has been supported by the National Natural Science Foundation of China (Grant No. 12241101, 42174192, and 11872128) and the Pandeng Program of National Space Science Center, Chinese Academy of Sciences.

**Review statement.** This paper was edited by XX and reviewed by XX anonymous referees.

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
