# Peer review of "Observation and simulation of neutral air density in the middle atmosphere during the 2021 sudden stratospheric warming event"

_EGUsphere, 2023_

## Author Response (AR1)

**1. Response to reviewer 1**

The SSWs could cause the dramatic change in the temperature, winds and components in the middle atmosphere, which has been widely reported in literature. In contrast, the neutral density evolution during the SSW is less reported. However, the density is one of the most important parameters for the atmosphere drag force calculation in the design of aircraft design. This research reports on the density evolution during the middle atmosphere during the 2021 major SSW event using the observation and simulation. The satellites data show a rapid increase of over 50% in the mesospheric density at high latitudes around the onset date during the SSW. The global view shows that the influenced area extends to the middle latitudes. Beijing lidar observed obvious density disturbances during the SSW event. The simulation using SD-WACCM demonstrates that the observed enhanced density is primarily attributed to the altered planetary waves and residual circulation during the SSW event. The density variation at altitude layers during the SSW is different from a simple inference by the temperature at certain pressure levels. This study is interesting and the results are sound. However, there are still some very minor concerns that should be considered. Therefore, I recommend a minor revision.

We thank you for your review and helpful comments.

Major comments:

The authors emphasize the importance of the BD circulation for the density change during the SSW, and say that the BDC transport denser air from lower latitudes to higher latitudes. However, the climatological distribution of the air density is not shown in the paper, which is a core plot. Based on Figure 3, I do not see very clearly the contrast of the air density between lower and higher latitudes. Only showing the pressure, the key message is not clearly present.

In the original manuscript, Figure 6d has shown that the temporal evolution of atmospheric density at 56 km. The density over the equator is higher than density over high-latitude in December before the onset, while the density over the equator is higher than density over the equator during the SSW. However, since this figure is a simulation result, we adopt the reviewer's suggestion to add a global distribution corresponding in the revised manuscript, which gives the relative deviation of the global density with the global mean density (Figure 1). In the climatological situation, the density over the equator is higher than the density of the other regions at 20-25 km, and the density over the Southern Hemisphere is overall higher than the density over the Northern Hemisphere at 25-90 km. The global distribution on 20 December 2020 is generally consistent with climatic winter feature. On 4 January 2021, the atmospheric density over the Arctic region at 30-85 km has increased by more than 20% larger than the global mean density, and this increase is relatively significant compared to the climatic Standard deviations. On 7 January 2021, the atmospheric density over the Arctic region at 50-90 km quickly returns to that situation before the onset, while atmospheric density at the 30-50 km is still higher. The maximum increase located at 46 km is also more significant compared to the climatic Standard deviations. This added figure gives the contrast of the air density between lower and higher latitudes. The Figure 2 in the original manuscript demonstrates the contrast between the air density distributions during the SSW and the climatic Standard deviations.

[Figure]

**Figure 1: Relative deviations between the zonal mean density and the global mean density: (a) November 20, 2020; (b) January 4, 2021; and (c) January 7, 2021 from Aura/MLS data and (d) climatic winter average obtained for the period from 2004 to 2021.**

Some jargons should be defined in the main text or in the figure captions. The authors claim that they show the deviation for the air density relative to the climatology. But the legend shows the relative change with a unit %. So where can the readers find the definition for the air density deviation (or relative deviation)?

Since the atmospheric density decreases exponentially with altitude, relative density deviation was used to represent the density change. We have added the definition for the relative density deviation $\delta$ at each altitude as:

$$\delta = \frac{\rho_o - \rho_r}{\rho_r} \times 100\%$$

where $\rho_o$ is the observed density and $\rho_r$ is the reference density. We have also revised the other similar details.

The authors only emphasize the dynamics importance for the change of the air density. But actually, the ideal state equation also explains the change of the air density in some circumstances. P=rho*R*T. You might find that the P and the T increase suddenly during the SSW, so which increases faster? If the P increases faster than the T, the increase in rho is also true, and the ideal state equation can also explain the change in the air density. I suggest to include a comparison between the thermodynamics and dynamics.

In the original manuscript, we have discussed that the temperature cooling magnitude of only approximately 20 K (8%) at 58 km could not cause the density enhancement of more than 50% recorded in the observational data. Figure 2 show the temporal evolution of the temperature and density at the isobaric levels and the corresponding approximate height is indicated in the right ordinate. This isobaric-level form temperature result is commonly given at present, such as Figure 1 in Manney et al. (2009), Figure 1 in Chandran and Collins (2014), Figure 4 in Kodera et al. (2016) and Figure1 in Lu et al. (2021). The accurate density information is hardly Most previous studies only give the atmospheric temperature change (Figure 2a), not the density and pressure, which unable to provide accurate density information to the aircraft designers and the aircraft industry. Since the atmospheric drag calculated on flight experiments should considers the density at a given altitude, the altitude results in our work are very meaningful. This large density increase should cause more attention in the calculation of atmospheric drag. The above discussion will be added in the revised manuscript.

[Figure]

**Figure 2: Temporal evolution of the temperature and density from December 1, 2020, to February 28, 2021. (a) Zonal temperature and (d) the density deviation between that during the SSW event and the climatology over 70° N from the MLS data. The vertical black lines indicate the onset date.**

Minor comments:

Line 23: Please be more specific for the "upper air model". I do not understand.

The "upper air model" means the reference atmosphere model used in the design of aircraft (Hale et al., 2002). We have added a description in the revised manuscript, as "standard upper air models, such as the 1976 US reference atmosphere".

Line 32: These references are too old and some most recent publications for the Southern Hemisphere SSW should be mentioned (doi: 10.1029/2020JD032723)

Thank you for the useful recommended manuscript. It is added in the revised manuscript.

Line 54: Please clarify the necessity of the density investigation.

It is reported that a dramatically varying actual atmospheric density often contributes to a negative angle of attack bias, which can have adverse thermal consequences.

Line 98: When introducing Beijing lidar, consider providing the detection principle or method.

We have added the detection principle: "The lidar emits pulsed laser, which elastically collides with atmospheric molecules. The received backward Rayleigh scattering signal can be captured by the detector of the lidar system and is used to calculate the atmospheric density and temperature."

Figure 3: The negative values are all too weak. Please modify the legend colors to match the figure well.

Thank you for your advice. The Figure 3 has been modified in the revised manuscript.

Line 230: Revised the sentence "the lidar density data constitute a directed parameter".

We have revised this sentence: "Furthermore, as the lidar density data is a directed parameter by Beijing lidar system, the consistency between lidar observations with the Aura/MLS observations confirms the computational methods described in Sect. 2. "

Figure 5: Here is not the PW1/2, but the wave amplitude? (see Line 234)

The reviewer is right. We have revised it.

Line 267: I can not see the climatological density contrast between the lower latitudes and higher latitudes.

We have added a global climatological density distribution (Figure 1d). In climatic winter

average situation, the density over the equator is higher than the density over higher latitudes at 20-25 km, and the density over the Southern Hemisphere is overall higher than the density over the Northern Hemisphere at 25-90 km. The feature on 20 December 2020 is generally consistent with the climatology.

Line 300: The sharp increase in air density is observed during the 2021 SSW. Whether the similar variations have occurred during other SSWs and what is the difference?

According to the classification based on the stratospheric response (Kodera et al., 2016), 2022 major SSW is more of a mixed type of warming event. We investigated two other typical style SSW. The 2006 major SSW was a typical displacement type event triggered by the PW1 (Chandran and Collins, 2014), while the 2018 major SSW was a split type event by PW2 (Rao et al., 2018). The sudden positive anomalies of neutral air density are also observed around the onset date in the 2018 and 2006 major SSW. In 2018, the maximum value of 49.54% at 48 km occurs on 15 February lags 4 days after the onset date. In 2006, the maximum value of 51.44% at 61 km occurs on 11 January, 11 days before the onset date.

References

Chandran, A. and Collins, R.: Stratospheric sudden warming effects on winds and temperature in the middle atmosphere at middle and low latitudes: a study using WACCM, 32, 859-874, https://doi.org/10.5194/angeo-32-859-2014, 2014.

Hale, N., Lamotte, N., and Garner, T.: Operational Experience with Hypersonic Entry of the Space Shuttle, AIAA/AAAF 11th International Space Planes and Hypersonic Systems and Technologies Conference, Orleans, France,    https://doi.org/10.2514/6.2002-5259,

Kodera, K., Mukougawa, H., Maury, P., Ueda, M., and Claud, C.: Absorbing and reflecting sudden stratospheric warming events and their relationship with tropospheric circulation, 121, 80-94, https://doi.org/10.1002/2015JD023359, 2016.

Lu, Q., Rao, J., Liang, Z., Guo, D., Luo, J., Liu, S., Wang, C., and Wang, T.: The sudden stratospheric warming in January 2021, J Environmental Research Letters, 16, 084029, https://doi.org/10.1088/1748-9326/ac12f4, 2021.

Manney, G. L., Harwood, R. S., MacKenzie, I. A., Minschwaner, K., Allen, D. R., Santee, M. L., Walker, K. A., Hegglin, M. I., Lambert, A., Pumphrey, H. C., Bernath, P. F., Boone, C. D., Schwartz, M. J., Livesey, N. J., Daffer, W. H., and Fuller, R. A.: Satellite observations and modeling of transport in the upper troposphere through the lower mesosphere during the 2006 major stratospheric sudden warming, Atmos. Chem. Phys., 9, 4775-4795, https://doi.org/10.5194/acp-9-4775-2009, 2009.

Rao, J., Ren, R., Chen, H., Yu, Y., and Zhou, Y.: The Stratospheric Sudden Warming Event in February 2018 and its Prediction by a Climate System Model, Journal of Geophysical Research: Atmospheres, 123, 13,332-313,345, https://doi.org/10.1029/2018JD028908, 2018.

**2.   Response to reviewer 2**

The present study explores the spatial and temporal distributions of (neutral) air density in the 2021 sudden stratospheric warming event using observational data (AURA/MLS, GNSS-RO and lidar) and global modeling whose dynamics is constrained using the reanalysis (SD-WACCM).

We thank you for your review and comments.

This study emphasizes the importance of density distributions in association with the evolution of anticyclonic and cyclonic vortices during the SSW event. However, flow evolution during the SSW event has been extensively studied using the concept of potential vorticity (e.g,, Harvey et al. 2002; Greer et

al. 2013; Lu et al. 2021). Potential vorticity is the dynamic variable that combines the thermodynamic process, and it has good property that it is conserved following fluid elements in the absence of heat and dissipation. That is, there is already good and well-known material invariant quantity that can be used for studies of polar vortex evolutions and mass transport around vortices. Hence, reviewer is not convinced about why we need air density as another key physical quantity for better understanding of vortex evolution.

The reviewer is right that potential vorticity is one of key variables and has been extensively studied. Nevertheless, air density is most key atmosphere quantity for the atmosphere drag force calculation in the design of aircraft (Weaver et al., 2011; Hale et al., 2002). One consequence of the density shears and resulting drag excursions is to cause frequent fluctuations in the angle of attack profile. These large fluctuations could aggravate the heating and also contribute to the saturation of the angle of attack at its corridor limits during maneuvers, which could deteriorate the drag control and the ranging accuracy. This may be the barrier between industry and the atmosphere community. Hale et al. (2002) also reported the model predicts significantly larger atmosphere dispersions at higher latitudes in the winter. SSW events, as the most spectacular global atmospheric phenomenon, were investigated that effects on temperature, wind, vortices and so on. However, as far as we know, there have been no reports of atmospheric density during SSW. The purpose of our work is to present the changes in density at altitude levels during SSW and to find the reasons. These results demonstrate a rapid enhanced density by 50% during the 2021 major SSW and is primarily attributed to the altered planetary waves and residual circulation during the SSW event. Our work is considered to promote the collaborations between industry and the atmosphere community.

As authors discussed, mass transport is important in mean-flow evolution associated with planetary waves, and the mass circulation can be approximately described by the residual circulation. Importance of mass circulation in polar vortex dynamics is not new, and the importance of "mass" transport would not require the use of air density as an additional diagnostic quantity. Reviewer agrees that the importance of neutral air density in the altitudes ($z = 200$-$1000$ km) of (Very) Low-Earth Orbit satellites in terms of air drag, but the topic of this study is the middle atmospheric phenomena in which substantial amount of mass transport would be due to radiative heating/cooling, planetary waves and gravity waves. In this sense, referring low-thermospheric studies like Oberhide et al. (2020) is not appropriate. Tidal waves rather than planetary waves would contribute more to mass circulation in the lower thermosphere.

As mentioned above, our study in density is requirement for the atmosphere drag force calculation in the design of aircraft. The atmosphere drag is a direct function of the neutral atmospheric density at a given altitude. The density evolution during SSW events is rarely reported. Dynamic diagnostics and mass transports in this paper can effectively explain the rapid increase in observed atmospheric density during SSW. This isobaric-level form result is commonly used at present, such as Figure 1 in Manney et al. (2009), Figure 1 in Chandran and Collins (2014), Figure 4 in Kodera et al. (2016) and Figure1 in Lu et al. (2021). The accurate density information at a given altitude is hardly speculated from previous studies. In this mean, our results are new and very meaningful. We have added above discussion in the revised manuscript.

The near space vehicles flight at the airspace 20–100 km and the space shuttle re-enter from an altitude of 122 km to the ground (Hale et al., 2002; Chen et al., 2023). Hale et al. (2002) has reported the density variations will highly influence engine performance, specific fuel consumption, drag, and flight control.

So, atmospheric density is not only important for Low-Earth Orbit satellites at the altitudes (z = 160-1600 km), but also for the vehicles in the middle atmosphere at the altitudes (z = 20-100 km). The latter is the topic of our study. We have introduced the importance in the first paragraph. Maybe it wasn't expressed clearly enough. So, we have revised the first paragraph.

"For the near space vehicles flighting at the airspace 20–100 km and the space shuttle, entry phase of which begins at an altitude of 122 km and ends at the ground, the atmosphere variations will highly influence specific fuel consumption, engine performance, drag, communication and flight control (Weaver et al., 2011; Chen et al., 2023). The density variations are manifested as density shear, differences from the standard atmosphere model, and density perturbations dependence on longitude, latitude, and season (Hale et al., 2002). The significant difference between the actual atmosphere density and widely used standard upper air models, such as the 1976 US reference atmosphere, has often been found and contributed to attack angle bias from the reference angle of attack profile, causing adverse thermal consequences (Champion, 1990; Hale et al., 2002). On several flight experiments, the atmospheric drag, a direct function of the neutral atmospheric density at a given altitude, has varied by up to 19% over a few seconds. It is indicated that middle atmospheric density variations require additional attention from aircraft designers and the aircraft industry (Hale et al., 2002; Weaver et al., 2011). Furthermore, 38 Starlink satellites were destroyed by a unexpected geomagnetic storm that led to a density enhancement of over 20% at ~210 km and a larger atmospheric drag on February 4, 2022 (Dang et al., 2022). Atmospheric density is not only important for Low-Earth Orbit satellites at the altitudes (160-1600 km), but also for the vehicles in the middle atmosphere at the altitudes (20-100 km)."

We agree that tidal waves rather than planetary waves would contribute more to mass circulation in the lower thermosphere. However, Oberhide et al. (2020) only the mentioned global-scale wave and did not distinguish the tidal waves and planetary waves. We cite their results here to illustrate the similarity in the material transport by SSW event, not the same waves. We have added a clarification based on your advice in the revised manuscript.

"However, it is should be noted that planetary waves are more dominant in the middle atmosphere, while tidal waves rather than planetary waves would contribute more in the lower thermosphere (Liu et al., 2010)"

Reviewer thinks this manuscript need to be rewritten such that this study either focus more on the middle atmospheric dynamics during the 2021 SSW in more meteorological context or focus more on the lower thermospheric impacts of the 2021 SSW using upper atmospheric observations and SD-WACCM-X. For this reason, reviewer would not recommend this manuscript for publication to ACP, although authors made significant efforts for comprehensive and quantitative analysis.

Thanks to the reviewer's comments. Our focus is the middle atmosphere during the 2021 SSW and the whole paper is centred on this purpose. Maybe, the two lower thermosphere papers cited here have confused the reviewers. We have presented a clear description of the need for middle atmosphere density studies and their value to the industry in the revised manuscript based on the above replies.

References

Champion, K. S. W.: Middle atmosphere density data and comparison with models, Advances in Space Research, 10, 17-26, https://doi.org/10.1016/0273-1177(90)90232-O, 1990.

Chandran, A. and Collins, R.: Stratospheric sudden warming effects on winds and temperature in the middle atmosphere at middle and low latitudes: a study using WACCM, 32, 859-874, https://doi.org/10.5194/angeo-32-859-2014, 2014.

Chen, B., Sheng, Z., and He, Y.: High-Precision and Fast Prediction of Regional Wind Fields in Near Space Using Neural-Network Approximation of Operators, Geophysical Research Letters, 50, e2023GL106115, https://doi.org/10.1029/2023GL106115, 2023.

Dang, T., Li, X., Luo, B., Li, R., Zhang, B., Pham, K., Ren, D., Chen, X., Lei, J., and Wang, Y.: Unveiling the Space Weather During the Starlink Satellites Destruction Event on 4 February 2022, Space Weather, 20, e2022SW003152, https://doi.org/10.1029/2022SW003152, 2022.

Hale, N., Lamotte, N., and Garner, T.: Operational Experience with Hypersonic Entry of the Space Shuttle, AIAA/AAAF 11th International Space Planes and Hypersonic Systems and Technologies Conference, Orleans, France, https://doi.org/10.2514/6.2002-5259,

Kodera, K., Mukougawa, H., Maury, P., Ueda, M., and Claud, C.: Absorbing and reflecting sudden stratospheric warming events and their relationship with tropospheric circulation, 121, 80-94, https://doi.org/10.1002/2015JD023359, 2016.

Liu, H. L., Foster, B. T., Hagan, M. E., McInerney, J. M., Maute, A., Qian, L., Richmond, A. D., Roble, R. G., Solomon, S. C., Garcia, R. R., Kinnison, D., Marsh, D. R., Smith, A. K., Richter, J., Sassi, F., and Oberheide, J.: Thermosphere extension of the Whole Atmosphere Community Climate Model, Journal of Geophysical Research: Space Physics, 115, https://doi.org/10.1029/2010JA015586, 2010.

Lu, Q., Rao, J., Liang, Z., Guo, D., Luo, J., Liu, S., Wang, C., and Wang, T.: The sudden stratospheric warming in January 2021, J Environmental Research Letters, 16, 084029, https://doi.org/10.1088/1748-9326/ac12f4, 2021.

Manney, G. L., Harwood, R. S., MacKenzie, I. A., Minschwaner, K., Allen, D. R., Santee, M. L., Walker, K. A., Hegglin, M. I., Lambert, A., Pumphrey, H. C., Bernath, P. F., Boone, C. D., Schwartz, M. J., Livesey, N. J., Daffer, W. H., and Fuller, R. A.: Satellite observations and modeling of transport in the upper troposphere through the lower mesosphere during the 2006 major stratospheric sudden warming, Atmos. Chem. Phys., 9, 4775-4795, https://doi.org/10.5194/acp-9-4775-2009, 2009.

Weaver, A. B., Alexeenko, A. A., Greendyke, R. B., and Camberos, J. A.: Flowfield uncertainty analysis for hypersonic computational fluid dynamics simulations, Journal of thermophysics heat transfer, 25, 10-20, https://doi.org/10.2514/1.49522, 2011.

**3. The list of all relevant changes made in the manuscript**

(1) We have changed the first paragraph of introduction in the original manuscript from "In the design of aircraft and when planning trajectories and forecasting flight conditions, standard upper air models are widely used (Champion, 1990). However, these models can only generate smooth climatic density profiles as a function of altitude and may include significant deviations from the natural atmosphere. Consequently, the difference between the actual drag and the reference design drag, a direct function of the neutral atmospheric density at a given altitude, has varied by up to 19% over a few seconds on several flight experiments (Hale et al., 2002). Furthermore, 38 Starlink satellites were destroyed by a unexpected geomagnetic storm that led to a density enhancement of over 20% at ~210 km and a larger atmospheric drag on February 4, 2022 (Dang et al., 2022). These events

indicate that such density variations require additional attention from aircraft designers and the aircraft industry (Weaver et al., 2011)." to "For the near space vehicles flighting at the airspace 20–100 km and the space shuttle, entry phase of which begins at an altitude of 122 km and ends at the ground, the atmosphere variations will highly influence specific fuel consumption, engine performance, drag, communication and flight control (Weaver et al., 2011; Chen et al., 2023). The density variations are manifested as density shear, differences from the standard atmosphere model, and density perturbations dependence on longitude, latitude, and season (Hale et al., 2002). The significant difference between the actual atmosphere density and widely used standard upper air models, such as the 1976 US reference atmosphere, has often been found and contributed to attack angle bias from the reference angle of attack profile, causing adverse thermal consequences (Champion, 1990; Hale et al., 2002). On several flight experiments, the atmospheric drag, a direct function of the neutral atmospheric density at a given altitude, has varied by up to 19% over a few seconds. It is indicated that middle atmospheric density variations require additional attention from aircraft designers and the aircraft industry (Hale et al., 2002; Weaver et al., 2011). Furthermore, 38 Starlink satellites were destroyed by a unexpected geomagnetic storm that led to a density enhancement of over 20% at ~210 km and a larger atmospheric drag on February 4, 2022 (Dang et al., 2022). Atmospheric density is not only important for Low-Earth Orbit satellites at the altitudes (160-1600 km), but also for the vehicles in the middle atmosphere at the altitudes (20-100 km)."

(2)   We have added the the definition for the relative density deviation in Lines 103-107 as:

"Since the atmospheric density decreases exponentially with altitude, relative density deviation was used to represent the density change. The relative density deviation $\delta$ at each altitude is calculated as:

$$\delta = \frac{\rho_o - \rho_r}{\rho_r} \times 100\%, \tag{6}$$

where $\rho_o$ is the observed density and $\rho_r$ is the reference density, which may differs in different figures and will be described in each figure."

(3)   We have changed the description of the Ground dataset: Beijing Rayleigh lidar from "The lidar deployed in Beijing (40.3° N, 116.2° E) is part of the lidar chain in the Meridian Project, which is a space environment monitoring system measuring the space environment from the ground (Wang, 2010). This lidar obtains the temperature and density profiles from 30 km to 70 km." to "The Rayleigh lidar deployed in Beijing (40.3° N, 116.2° E) is part of the lidar chain in the Meridian Project, which is a space environment monitoring system measuring the space environment from the ground (Wang, 2010). The lidar emits pulsed laser, which elastically collides with atmospheric molecules. The received backward Rayleigh scattering signal can be captured by the detector of the lidar system and is used to calculate the atmospheric density and temperature (Yue et al., 2014). In this paper, the temperature and density profiles from 30 km to 70 km were retrieved from lidar observational data from December 2020 to February 2021."

(4)   We have added a global density distribution figure and related analysis from line 181 to line 192 in the revised manuscript as:

"According to the observed temporal evolution of the density, the latitude–altitude density structures were investigated using the Aura/MLS measurements for three different stages of the 2021 SSW event. Figure 2 gives the relative deviation of the global density with the global mean density for November 20, 2020 (pre-SSW); January 4, 2021 (around the onset date); and February 1, 2021 (after the onset date). The climatological mean density were obtained from the 18-year observational data of Aura/MLS during the period of 2004–2021. In the climatological situation, the density over the equator is higher than the density of the other regions at 20-25 km, and the density over the Southern Hemisphere is overall higher than the density over the Northern Hemisphere at 25-90 km. The global distribution

on 20 December 2020 is generally consistent with climatic winter feature. On 4 January 2021, the atmospheric density over the Arctic region at 30-85 km has increased by more than 20% larger than the global mean density, and this increase is relatively significant compared to the climatic Standard deviations. On 7 January 2021, the atmospheric density over the Arctic region at 50-90 km quickly returns to that situation before the onset, while atmospheric density at the 30-50 km is still higher. The maximum increase located at 46 km is also more significant compared to the climatic Standard deviations. "

[Figure]

**Figure 2: Relative deviations between the zonal mean density and the global mean density: (a) November 20, 2020; (b) January 4, 2021; and (c) January 7, 2021 from Aura/MLS data and (d) climatic winter average obtained for the period from 2004 to 2021.**

(5)  We have modified the number of the Fig. 2-8 to the number 3-9.

(6)  We have made a change from line 203 to line 204 in the revised manuscript as: "To test whether the density change during the 2021 major SSW is significant, the relative density deviations from the climatological mean situation are compared with the climatological standard deviation shown in Fig. 3."

(7)  We have modified the legend colors of Fig.3 to match the figure well.

[Figure]

**Figure 3: Relative deviations between the zonal mean density and the climatological average density: (a) November 20, 2020; (b) January 4, 2021; and (c) January 7, 2021 from Aura/MLS data. (d) The standard deviations of the zonal mean density for the winter average obtained for the period from 2004 to 2021.**

(8) We have changed that sentence "In addition, the cooling magnitude of approximately 20 K (8%) at 58 km cannot lead to a density enhancement of above 50%." to "In addition, the cooling magnitude of approximately 20 K (roughly -8% of pre-existing temperature) at 58 km cannot lead to a density enhancement of above 50%, assuming other conditions are stable." from line 265 to line 267 in the revised manuscript.

(9) We have modified the caption of Fig. 6 from "Temporal evolutions of the (a) temperature, (b) relative density deviation, (c) planetary wavenumber 1 (PW1), and (d) planetary wavenumber 2 (PW2) of the pressure over 70° N simulated by SD-WACCM. The vertical black lines indicate the onset date, and the dashed lines in panels (a) and (b) indicate the maximum temperatures as the stratopause altitude." to "Temporal evolutions over 70° N simulated by SD-WACCM: (a) The zonal mean temperature. (b) The zonal mean density. (c and d) The amplitudes of planetary wavenumber 1 (PW1), and planetary wavenumber 2 (PW2) of the pressure. The vertical black lines indicate the onset date, and the dashed lines in panels (a) and (b) indicate the maximum temperatures as the stratopause altitude."

(10) We have added a comparison between the thermodynamics and dynamics in Sect.4 (Discussion and conclusions) in the revised manuscript from line 337 to line 347 as "Figure 10 show the temporal evolution of the temperature and density at the isobaric levels and the corresponding approximate height is indicated in the right ordinate. This isobaric-level form result is commonly used at present (Manney et al., 2009; Chandran and Collins, 2014; Kodera et al., 2016; Lu et al., 2021). It can be seen from the Fig. 10 that the stratospheric temperature warming around the onset date causes the density to decrease by -14.33% at ~ 43km, and the mesospheric temperature cooling causes the density to increase by 14.42% at 75 km. This density change at the isobaric levels can be considered as the results caused by the change in thermodynamic temperature change, which is under the same pressure. Supposing that the atmosphere is stationary, the temperature cooling magnitude of only approximately 20 K (-8%) at 58 km could not cause the density enhancement of more than 50% recorded in the observational data. These huge differences between this density evolution at the isobaric levels (Fig. 10) and the observation density evolution at altitudes (Fig. 1), indicate that the density increment as high as 50% around the onset data is mainly caused by planetary waves (Fig. 6 and 7)." and from line 337 to line 347 as "On the other hand, most previous studies only give the atmospheric temperature change like Fig. 10a, not the density and pressure, which unable to provide accurate density information to the aircraft designers and the aircraft industry. Since the atmospheric drag calculated on flight experiments should considers the density at a given altitude, the altitude results in our work are very meaningful. This large density increase should cause more attention in the calculation of atmospheric drag."

(11) We have added Fig.10 in the revised manuscript.

[Figure]

**Figure 10: Temporal evolution of the temperature and density from December 1, 2020, to February 28, 2021. (a) Zonal temperature and (d) the density deviation between that during the SSW event and the climatology over 70° N from the MLS data. The vertical black lines indicate the onset date.**

(12) We have changed that sentence "The changes in the RMC around the onset seen in Fig. 8 last much longer than the other changes." to "The changes in the RMC around the onset seen in Fig. 8 last much longer than the other changes at other times." from line 364 to line 365 in the revised manuscript.

(13) We have added a sentence "However, it is should be noted that planetary waves are more dominant in the middle atmosphere, while tidal waves rather than planetary waves would contribute more in the lower thermosphere (Liu et al., 2010)." from line 375 to line 376 in the revised manuscript.

(14) We have added density variations during other SSWs at the end of Sect.4 (Discussion and conclusions) as "According to the classification based on the stratospheric response (Kodera et al., 2016), 2022 major SSW is more of a mixed type of warming event. We have also investigated two other typical style SSW. The 2006 major SSW was a typical displacement type event triggered by the PW1 (Chandran and Collins, 2014), while the 2018 major SSW was a split type event by PW2 (Rao et al., 2018). The sudden positive anomalies of neutral air density are also observed around the onset date in the 2018 and 2006 major SSW. In 2018, the maximum value of 49.54% at 48 km occurs on 15 February lags 4 days after the onset date. In 2006, the maximum value of 51.44% at 61 km occurs on 11 January, 11 days before the onset date. Their difference detail needs more explorations in future work."